# Deformation Potentials: Towards a Systematic Way beyond the Atomic Fragment Approach in Orbital-Free Density Functional Theory

**DOI:** 10.3390/molecules26061539

**Published:** 2021-03-11

**Authors:** Kati Finzel

**Affiliations:** Faculty of Chemistry and Food Chemistry, Technische Universität Dresden, Bergstraße 66c, 01069 Dresden, Germany; Kati.Finzel@tu-dresden.de

**Keywords:** orbital-free density functional theory, bifunctional approach, Pauli potential, Pauli kinetic energy, chemical bonding, real space, deformation potentials

## Abstract

This work presents a method to move beyond the recently introduced atomic fragment approximation. Like the bare atomic fragment approach, the new method is an ab initio, parameter-free, orbital-free implementation of density functional theory based on the bifunctional formalism that treats the potential and the electron density as two separate variables, and provides access to the Kohn–Sham Pauli kinetic energy for an appropriately chosen Pauli potential. In the present ansatz, the molecular Pauli potential is approximated by the sum of the bare atomic fragment approach, and a so-called deformation potential that takes the interaction between the atoms into account. It is shown that this model can reproduce the bond-length contraction due to multiple bonding within the list of second-row homonuclear dimers. The present model only relies on the electron densities of the participating atoms, which themselves are represented by a simple monopole expansion. Thus, the bond-length contraction can be rationalized without referring to the angular quantum numbers of the participating atoms.

## 1. Introduction

The Hohenberg–Kohn theorems [1] lay the foundation for a purely density-based description of quantum mechanics, covering, in principle, all aspects of electronic structure theory and, consequently, chemical bonding. However, the deficiencies of orbital-free density functional approaches, such as the failure to reproduce the atomic shell structure [2,3] and the lack of proper chemical bonding [4], are well known drawbacks [5]. As a consequence, orbital-free density functional theory (OF-DFT) was abandoned by most researchers and the widespread believe in the *native insufficience of OF-DFT* grew within the community. Given the enormous success of the orbital-based Kohn–Sham (KS) method [6] (exploiting a fictious system of non-interacting electrons in order to approximate the kinetic energy for the system of interest), where the burden of approximation is left to the exchange-correlation energy only, and where sophisticated methods have been developed over the years [7,8], its original, purely orbital-free variant fell into oblivion.

Noticeable exceptions to the recent work on OF-DFT have been provided by Nagy and coworkers [9,10], directly addressing the Pauli kinetic energy and the Pauli potential, a concept that was derived by March [11] in 1986 by separating the total kinetic energy into an analytically known expression, namely, the von Weizsäcker kinetic energy [12], and an unknown remainder, called Pauli kinetic energy. Since then, Pauli kinetic energy and its potential have been subjected to various theoretical studies [11,13,14,15,16,17,18,19,20,21,22].

Valuable contributions, via the path of generalized-gradient-expansion techniques and in-depth investigations of parameterization techniques, have been provided by Trickey et al. [4,23,24]. To date, expansion techniques remain the most common research line in the field of OF-DFT [3,12,25,26,27,28,29,30,31,32,33,34,35,36,37,38,39,40,41,42,43]. Surely, one of the highly celebrated benefits of OF-DFT is its enormous gain in computational speed-up, as shown by Carter et al. [44,45,46,47] and related groups [48,49], but also the interest in conceptual insights has recently been renewed [50,51].

The aim of the present work is to provide a further contribution in this conceptual direction. It has recently been shown that chemical bonding of increasing accuracy can be obtained from non-parameterized, ab initio, orbital-free methods [52,53,54] via the bifunctional approach [52,53,54,55,56,57] (exploiting the homogenous scaling behavior of the functionals [58]), and the introduced atomic fragment approximation [52,53,54,56,57]. The target of the present study is to shed some light on potential systematic pathways beyond the atomic fragment approximation. The paper contains a compilation of bond lengths for second-row homonuclear dimers using the bare atomic fragment approach, showing the ability and limitations of this ansatz. Based on a detailed investigation of the orbital-based KS Pauli potential for the respective dimers, with the inclusion of ideas from traditional molecular orbital (MO) theory and their influence on the respective electronic structure graphs of molecular Pauli potential, a new form of the approximate molecular Pauli potential is proposed. The new model is based on the idea of the constructive versus destructive interaction of atomic electron densities, and its performance with respect to equilibrium bond lengths is tested accordingly. The final task is to reproduce traditional chemical concepts [59,60,61] such as multiple bonding from a purely density-based ansatz.

## 2. Theory

Within an OF-DFT method, the Pauli kinetic energy $T_{P}$ remains the only unknown functional expression, while approximations for the exchange-correlation part, resulting from the electron–electron repulsion, are generously accepted as sufficiently well-described by a local theory level (LDA) in order to focus on the more problematic issue of representing the kinetic energy of the system. As for the well-known KS-DFT method, the foundations of OF-DFT lie in the Hohenberg–Kohn theorems [1], according to which the total electronic energy *E* of a system can be expressed as a functional of the electron density ρ
(1)$$E\left[\rho \right]=T_{s}\left[\rho \right]+V_{ee}\left[\rho \right]+V_{Z}\left[\rho \right].$$
Here, $T_{s}\left[\rho \right]$ is the non-interacting kinetic energy, $V_{ee}\left[\rho \right]$ is the Coulomb repulsion between the electrons, and $V_{Z}\left[\rho \right]$ is the electron–nuclear attraction energy. The latter term is known exactly as electron density functional
(2)$$V_{Z}\left[\rho \right]=\int \rho \left(\vec{r}\right)v_{Z}\left(\vec{r}\right)\;d\vec{r}\;\;$$
where the nuclear potential of a molecule $v_{Z}\left(\vec{r}\right)=\sum_{A}v_{Z}^{A}\left(\vec{r}\right)$ is given by the superposition of all atomic nuclear potentials $v_{Z}^{A}\left(\vec{r}\right)=-Z_{A}/\left|\vec{r}-{\vec{R}}_{A}\right|$, with $Z_{A}$ being the nuclear charge and ${\vec{R}}_{A}$ the nuclear coordinates. The electron–electron repulsion $V_{ee}\left[\rho \right]$ is usually split into the Hartree energy $E_{H}\left[\rho \right]$ and the exchange-correlation energy $E_{XC}\left[\rho \right]$, where the Hartree energy is given by
(3)$$E_{H}\left[\rho \right]=\frac{1}{2}\int \int \frac{\rho \left(\vec{r}\right)\rho \left({\vec{r}}^{\prime }\right)}{|\vec{r}-{\vec{r}}^{\prime }|}\;d{\vec{r}}^{\prime }\;d\vec{r}.$$
As pointed out in the beginning of this section, in OF-DFT methods, the focus is set on approximating the kinetic energy and thus, for simplicity reasons, the exchange-correlation part is expressed as local exchange energy only $E_{X}^{\mathrm{LDA}}\left[\rho \right]$ [1]
(4)$$\begin{array}{c} E_{X}^{\mathrm{LDA}}\left[\rho \right]=-C_{X}\int \rho^{\frac{4}{3}}\left(\vec{r}\right)\;d\vec{r}\;\; \end{array}$$
with $C_{X}=3/\left(4\pi \right){\left(3\pi^{2}\right)}^{1/3}\approx 0.73856$.

Following the idea of March, the non-interacting kinetic energy $T_{s}\left[\rho \right]$ can be regarded as being constructed from a bosonic part, the von Weizsäcker term [12] $T_{W}$, and a remainder, called Pauli kinetic energy $T_{P}$, which, consequently, is defined as the difference [11]
(5)$$T_{P}=T_{s}-T_{W}.$$
The von Weizsäcker kinetic energy $T_{W}$ is given analytically in terms of the electron density [11,12]
(6)$$T_{W}\left[\rho \right]=\int t_{W}\left(\vec{r}\right)\;d\vec{r}=\int \frac{1}{8}\frac{{\left[\vec{\nabla }\rho \left(\vec{r}\right)\right]}^{2}}{\rho (\vec{r})}\;d\vec{r},$$
where the integral kernel $t_{W}\left(\vec{r}\right)$ is called Weizsäcker kinetic energy density. Finally, the Pauli kinetic energy $T_{P}$ remains the only unknown functional expression for a purely orbital-free description of quantum mechanics. In recent publications [52,53], it was shown that the Pauli kinetic energy can be evaluated from the so-called bifunctional expression
(7)$$T_{P}\left[\rho ,v_{P}\right]=-\frac{1}{2}\int \rho \left(\vec{r}\right)\;\vec{r}\cdot \vec{\nabla }v_{P}\left(\vec{r}\right)\;d\vec{r}\;\;\;$$
involving both the electron density $\rho (\vec{r})$ and the Pauli potential $v_{P}\left(\vec{r}\right)$ as two separate variables. The bifunctional expression exploits the homogenous scaling behavior and the corresponding formulas [58] of the otherwise unknown functional expression. Consequently, the bifunctional expression yields exactly the KS Pauli kinetic energy, when the molecular electron density and the molecular Pauli potential are inserted into Equation (Equation 7). Of course, the Pauli potential of the molecule is only known in terms of the molecular KS eigenfunctions $\phi_{i}\left(\vec{r}\right)$ and their respective eigenvalues $\epsilon_{i}$ [13]
(8)$$v_{P}\left(\vec{r}\right)=\frac{\delta T_{p}}{\delta \rho }=\frac{t_{P}\left(\vec{r}\right)}{\rho (\vec{r})}+\sum_{i}\left(\epsilon_{M}-\epsilon_{i}\right)\frac{|\phi_{i}\left(\vec{r}\right){|}^{2}}{\rho (\vec{r})}\;\;\;$$
where the sum runs over all occupied eigenfunctions, $\epsilon_{M}$ is the highest occupied eigenvalue of the system, and the Pauli kinetic energy $t_{P}\left(\vec{r}\right)$ is given by
(9)$$t_{P}\left(\vec{r}\right)=\frac{1}{2}\sum_{i}{\left|\nabla \phi_{i}\left(\vec{r}\right)\right|}^{2}-t_{W}\left(\vec{r}\right).$$

In an orbital-free formalism, the KS eigenfunctions and eigenvalues are, of course, not available, and, thus, sufficiently accurate approximations for the molecular Pauli potential have to be found. The recently introduced atomic fragment approximation can be seen as a somewhat natural first step in order to approximate the molecular entity. In the applied formalism, the choice of atomic fragments markedly influences the ability of the method to properly model chemical bonding curves. It has been shown that artificially constructed closed-shell atoms better mimic the corresponding atomic fragments within the molecule, and thus yield better equilibrium bond lengths compared to experimental data [54] than the ordinary ground-state atoms. In the recently introduced OF-DFT methods, the total kinetic energy was given by
(10)$$\begin{array}{c} E\left[\rho ,v_{P}^{\Omega }\right]=T_{W}\left[\rho \right]+T_{P}^{\Omega }\left[\rho ,v_{P}^{\Omega }\right]+E_{H}\left[\rho \right]+E_{X}^{\mathrm{LDA}}\left[\rho \right]+V_{Z}\left[\rho \right]. \end{array}$$
employing the fragment Pauli kinetic energy $T_{P}^{\Omega }\left[\rho ,v_{P}^{\Omega }\right]$ given in terms of a bifunctional
(11)$$T_{P}^{\Omega }\left[\rho ,v_{P}^{\Omega }\right]=-\frac{1}{2}\int \rho \left(\vec{r}\right)\;\vec{r}\cdot \vec{\nabla }v_{P}^{\Omega }\left(\vec{r}\right)\;d\vec{r}\;\;\;$$
where the molecular Pauli potential is approximated by its atomic fragment variant
(12)$$v_{P}^{\Omega }\left(\vec{r}\right)=\sum_{A}v_{P}^{A}\left(\vec{r}-{\vec{R}}_{A}\right)\;\;\;$$
and $v_{P}^{A}\left(\vec{r}\right)$ is the KS Pauli potential for atom *A*. Note that the approximations made here are due to physically meaningful considerations of the molecular formation process. At this level of theory, neither approximate analytical functional expression, nor parameters or fitting procedures have been introduced.

In this work, systematic improvements beyond the atomic fragment approximation are studied. As mentioned above, the bifunctional approach yields exactly the KS Pauli kinetic energy, when the Pauli potential in the bifunctional expression equals the KS Pauli potential. Consequently, the following ansatz, employing a deformation potential $v_{P}^{\mathrm{def}}\left(\vec{r}\right)$
(13)$$v_{P}^{\Omega \mathrm{def}}\left(\vec{r}\right)=v_{P}^{\Omega }\left(\vec{r}\right)+v_{P}^{\mathrm{def}}\left(\vec{r}\right)\;\;\;$$
can improve the bare atomic fragment approach when a sufficiently accurate approximation for $v_{P}^{\mathrm{def}}\left(\vec{r}\right)$ is inserted. The corresponding Pauli kinetic energy is evaluated again via the bifunctional formalism
(14)$$T_{P}^{\Omega \mathrm{def}}\left[\rho ,v_{P}^{\Omega \mathrm{def}}\right]=-\frac{1}{2}\int \rho \left(\vec{r}\right)\;\vec{r}\cdot \vec{\nabla }v_{P}^{\Omega \mathrm{def}}\left(\vec{r}\right)\;d\vec{r}\;\;\;$$
and the total electronic energy is finally given by
(15)$$\begin{array}{c} E\left[\rho ,v_{P}^{\Omega \mathrm{def}}\right]=T_{W}\left[\rho \right]+T_{P}^{\Omega \mathrm{def}}\left[\rho ,v_{P}^{\Omega \mathrm{def}}\right]+E_{H}\left[\rho \right]+E_{X}^{\mathrm{LDA}}\left[\rho \right]+V_{Z}\left[\rho \right]. \end{array}$$
While it is relatively easy to obtain the KS Pauli kinetic energy for a given system of interest, and, consequently, its total KS energy, e.g., simply by inverting the KS equations and defining the deformation potential as the difference between the KS Pauli potential and the bare atomic fragment approach, it is of great difficulty to predict the performance of a given approximate deformation potential in advance. The challenge here is not only to yield good energetic agreement for a special case, but to obtain reasonable energy differences, e.g., for a molecule at various geometries. The reasonable description of chemical bonding, thus lies in the choice of an appropriate deformation potential $v_{P}^{\mathrm{def}}\left(\vec{r}\right)$.

In this work, a parameter-free ansatz that relies on the constructive versus destructive interaction between the atomic fragments is investigated. As will be shown in the subsequent section, the dominant part of the KS Pauli potential, responsible for the deviation from the bare atomic fragment ansatz, is given by the first term in Equation (Equation 8). Therefore, the deformation potential is represented by the following ansatz
(16)$$v_{P}^{\mathrm{def}}\left(\vec{r}\right)=\frac{t_{P}^{\mathrm{def}}\left(\vec{r}\right)}{\rho (\vec{r})}$$
where the deformation Pauli kinetic energy $t_{P}^{\mathrm{def}}\left(\vec{r}\right)$ is constructed from those electrons which take part in the bonding. Here, the interaction between atoms *A* and *B* may be of a constructive or destructive nature
(17)$$\Phi_{\pm }\left(\vec{r}\right)=\frac{1}{\sqrt{1\pm S}}\left[\Phi_{A}\left(\vec{r}\right)\pm \Phi_{B}\left(\vec{r}\right)\right]$$
as indicated by the plus and minus sign, respectively, and *S* is the overlap between the functions $\Phi_{A}\left(\vec{r}\right)$ and $\Phi_{B}\left(\vec{r}\right)$. The individual atomic contributions $\Phi_{A}\left(\vec{r}\right)$ and $\Phi_{B}\left(\vec{r}\right)$ are evaluated with the help of the respective atomic shape functions [62]
(18)$$\Phi_{A}\left(\vec{r}\right)=\sqrt{\frac{\rho_{A}\left(\vec{r}\right)}{N_{A}}}\;\;\;$$
where $N_{A}$ and $\rho_{A}\left(\vec{r}\right)$ are the number of electrons and the electron density of atom *A*, respectively.

Thus, for the ansatz made in this work, it suffices to scale Equation (Equation 17) with the chosen number of constructive and destructive electronic interactions within the molecule and insert the corresponding formulas into Equations (Equation 9) and (Equation 16) together with the appropriately scaled von Weizsäcker term (describing the same number of electrons) in order to determine the deformation potential
(19)$$v_{P}^{\mathrm{def}}\left(\vec{r}\right)=\frac{c\frac{1}{2}{\left[\nabla \Phi_{+}\left(\vec{r}\right)\right]}^{2}+d\frac{1}{2}{\left[\nabla \Phi_{-}\left(\vec{r}\right)\right]}^{2}-\frac{c+d}{N}t_{W}\left(\vec{r}\right)}{\rho (\vec{r})}\;\;\;$$
where *c* and *d* are the number of constructive and destructive terms, respectively and *N* is the total number of electrons in the system. Note that this approach allows for the treatment of electron-counting rules, e.g., originating from an MO graph, within an OF-DFT method, as well as for the separate treatment of core and valence electrons.

The input of *c* and *d* does, of course, require some knowledge from electronic structure theory, and so does the concept of separation between core and valence electrons. Such concepts, however, are at the basis of almost every chemical reasoning and find applications outside quantum chemistry as well, e.g., the commonly used octet rule which serves to explain the vast majority of molecular compositions. Additionally, the reader will notice the close connection of Equation (Equation 17) to the ansatz for classical one-electron functions in molecular orbital theory [63], which is the underlying reasoning for the proposed model. However, there are two main differences rendering the present model an approximation of the orbital-based KS PP, cf. Equation (Equation 8). First of all, the ansatz given in Equation (Equation 17) is not considered an eigenfunction, and second, the whole deformation potential is constructed with the help of one single atomic function, namely, its electron density, whereas, in classical MO theory, the ansatz is split according to the atomic angular quantum numbers, both conceptually and computationally. Of course, the present model can be made exact (in the sense to match the KS energy) by searching for those functions and corresponding values ϵ, cf. Equations (Equation 8) and (Equation 9), obeying the necessary nodal conditions and become the KS eigenfunctions and eigenvalues at the end of the optimization process. In this case, however, one would finally have performed a classical KS calculation.

The point, here, is to investigate which parts of the molecular KS eigenfunctions are the necessary ingredients for an approximate PP in order to properly model chemical bonding. As will be shown in the following section, in case of chemical bonding, such a mandatory ingredient seems to be the proper mixture (according to the MO concept) of constructive and destructive terms given by Equation (Equation 17), as the impact of the nodal plane given by the destructive combination is responsible for the increasing Pauli repulsion in O${}_{2}$, F${}_{2}$, and Ne${}_{2}$. In other cases, like modeling the proper atomic shell structure for atoms in their groundstate, nodal planes do not seem to play an important role, but a proper relationship is needed between the exponential decay and the model for the eigenvalues [64].

## 3. Results and Discussion

Figure 1 compiles the Pauli potential (PP) from the bare atomic fragment approach, together with the molecular PP obtained from KS orbitals as well as its components, cf. Equation (Equation 8), for the second-row homonuclear dimers at their respective equilibrium bond distances obtained from KS/LDA/QZ4P calculations [65]. Here, the heaviest dimer, Ne${}_{2}$, is shown in the first row, followed by successively lighter dimers, until Li${}_{2}$ is shown in the last row. At first glance, the close similarity between the bare atomic fragment approach (first column) and the molecular KS Pauli potential (second column) can be noted. This is due to the fact that the PP exhibits its most dominant values within the core regions: those regions which are not affected during chemical processes. For this reason, the bare atomic fragment potential serves as a natural starting point for reasonable approximations of the molecular Pauli potential. However, as will be shown later, the bare atomic fragment approach yields systematically decreasing equilibrium bond length with the increasing nuclear charge of the participating atoms. While this is a favorable observation from Li${}_{2}$ to N${}_{2}$, the bare atomic fragment approach, thus, misses the reproduction of the increasing Pauli repulsion in O${}_{2}$, F${}_{2}$, and Ne${}_{2}$. The latter potentials are shown in the upper last rows of Figure 1. As can be seen from the figure, the atomic fragment potential deviates from the KS PP, especially in the bonding region. While the overall magnitude of the Pauli potential within this region is small, it exhibits a pronounced influence on the chemical bonding curve and, consequently, the equilibrium bond distance.

Compare the molecular PP (second column) with its components, $t_{P}\left(\vec{r}\right)/\rho \left(\vec{r}\right)$ and $\sum_{i}\left(\epsilon_{M}-\epsilon_{i}\right){\left|\phi_{i}\left(\vec{r}\right)\right|}^{2}/\rho \left(\vec{r}\right)$, shown in columns three and four, respectively. The latter mainly serves to increase the Pauli repulsion in the core regions, while the former takes part of the distortion in the bonding region, especially when molecular orbitals of antibonding character are filled up. Therefore, the deformation potential accounting for the various bonding scenarios was chosen to be constructed from the first part of Equation (Equation 8), while the differences in the eigenvalues are already contained in the bare atomic fragment potential. As mentioned in Section 2, the interaction between both atoms can be of either a constructive or destructive character, indicated by the plus and minus sign in Equation (Equation 17). In the case of molecular orbitals, the termini bonding and anti-bonding character would be appropriate. Note that only the destructive combination of atomic electron densities may increase the value of the PP at the bond-critical point (bcp), cf. Equation (Equation 17), since the contribution from the constructive interaction is always zero at the bcp. In principle, for each homonuclear dimer, one can investigate any combination of constructive versus destructive natures, unless the number of total electrons is not exceeded, but, of course, the most interest lies in those combinations that are in line with traditional electronic graphs from the molecular orbital theory [63,66].

In line with the ideas mentioned above, various deformation potentials have been created in order to test their performance with respect to chemical bonding. Here, adding constructive terms to the deformation potential should favor chemical bonding and, consequently, yield shorter equilibrium bond lengths, while adding destructive combinations should weaken the chemical bond and yield larger equilibrium bond distances or no bond at all. Exemplarily, the data are compiled for N${}_{2}$, see Table 1. The Table contains equilibrium bond distances for any combination of constructive versus destructive atomic interaction, unless the total number of electrons is exceeded. Since each N atom may, at most, contribute seven electrons, all possible combinations up to 14 electrons have been investigated. At first glance, most of the combinations yield unbound atoms. These are, of course, all the combinations that have more destructive than constructive terms, but also those combinations with an equal amount of constructive and destructive interactions (with the exception of the bare atomic fragment approach). From the chemical viewpoint, this is a favorable outcome. Remember that the present OF-DFT approach is based on traditional concepts from chemical bonding theory only, and does not contain any parameterization or data fitting. Of course, atoms that share more destructive than constructive interactions are unbound, and so are those with an equal amount, since, as shown by Kutzelnigg [63], the impact of an antibonding orbital in dimers is larger than the impact of the corresponding binding orbital, due to the normalization constant; see Equation (Equation 17). Note that, the combination of six constructive and four destructive, as well as the deformation potential constructed using eight constructive and six destructive terms also yield unbound atoms. Thus, within the present model, destructive terms are slightly oversized while the corresponding constructive terms cannot fully compensate for it. This is also reflected by the equilibrium bond length for the optimal MO scheme, in case of N${}_{2}$ being determined from the deformation potential with eight constructive and two destructive terms, in order to yield six electrons for the triple bond. The corresponding bond distance is, with 2.38 bohr, roughly 15% too large compared to its experimental value.

However, notice the systematic change in the equilibrium bond lengths for the proposed model. As mentioned in Section 2, obtaining a reasonable OF-DFT kinetic energy for a special case can be done numerically in a straightforward way by the inversion of the KS equations. Equally, one could think of a certain model with adjustable parameters in order to obtain the desired match (as was frequently done with the von Weizsäcker correction to the Thomas-Fermi theory [2]). In contrast to single-point matches, the design of kinetic energy functionals yielding appropriate energy differences, e.g., for varying nuclear coordinates, is still challenging. The model proposed in this work not only yields reasonable energy differences for a chosen deformation potential with a fixed number for *c* and *d*, thus yielding bound atoms with reasonable bond distances, it also yields reasonable results among different ansatzes (using different numbers for *c* and *d*). Changes in the bond length (the minima of the respective bonding curves) follow the reasoning of the traditional electronic structure theory. As can be seen from Table 1, increasing the number of constructive interactions for a given number of destructive terms decreases the corresponding bond length, while the inclusion of more destructive interaction terms (for a given number of constructive electron-sharing) systematically increases the respective bond length. Those findings apply to all examined test cases. As a consequence, the deformation potential for the corresponding MO scheme is expected to yield unbound atoms in the case of Be${}_{2}$ and Ne${}_{2}$, since, here, the number of constructive terms equals the number of destructive electron sharing, but F${}_{2}$ is expected to be unbound in this model, since the number of six destructive terms is rather high in this molecule and may not fully be compensated by the remaining eight constructive terms. Still, the question remains as to whether the proposed model is able to yield bond distances that are in accordance with our chemical understanding of single, double and triple bonds. N${}_{2}$ shall yield the shortest equilibrium bond distance within the list of the homonuclear second-row dimers, followed by its next nearest neighbors (X${}_{2}$, with X = C,O) and so on.

Equilibrium bond lengths from OF-DFT using the deformation potential, in accordance with the MO scheme, are compiled in Table 2. The model, however, does not seem to be suitable for Li${}_{2}$, in which case the calculations are not converging. As can be seen from the table, the abovementioned expectations for the model are met. Indeed, the present approach yields the shortest bond distance for N${}_{2}$ (exhibiting a classical triple bond) within the list of the second-row dimers. Note that the proposed model allows the examination of electron-counting rules based on a single function only, namely, the electron density of the participating atoms. The present approach, thus, allows for the rationalization of experimental findings, such as the bond-length contraction from B${}_{2}$ to N${}_{2}$, together with the subsequent widening from N${}_{2}$ to O${}_{2}$, based on the simple electronic scheme shown in Table 2. Note that no reference is made to the angular quantum number, since the electron density is given as a simple monopole expansion (see Section 3) in all cases. The present approach, thus, rationalizes the occurrence of multiple bonds and their respective bond-length contraction without introducing concepts like p-orbitals, which are usually employed in such discussions.

Finally, a comparison is made to bond distances from the orbital-based KS scheme, as well as to experimental bond lengths. Figure 2 shows the equilibrium bond distances for the second-row homonuclear dimers, evaluated with the ADF program [65] at LDA(Xonly) level using the QZ4P basis sets, shown in green, together with the corresponding experimental values [67,68], shown in black, and the bond lengths from the two recent OF-DFT approaches using the bare atomic fragment approximation, shown in blue, and the deformation potential, shown in red, respectively. As can be seen from the data, the bare atomic fragment approach misses the reproduction of the influence of multiple bonding. Since the model is a bare atomic fragment approach, the corresponding equilibrium bond lengths basically follow the size of the core regions [69] of the participating atoms. The bare atomic fragment approach accounts for the Pauli repulsion within each atom, as well as, to some extent, for the repulsion in between the atoms. Namely, the bare atomic fragment approach takes the repulsion due to the interaction of the incoming electron density from the approaching atom *B* with the Pauli potential at atom *A* into account. However, the influence on the molecular Pauli potential due to the interacting atoms is not captured by this ansatz, since the bare atomic fragment potential is not optimized during the molecular formation. The present work proposes an ansatz to relax the Pauli potential due to the influence of the other atoms, by the introduction of a deformation potential. The deformation potential is build from the electron density of the interacting atoms and can be of a constructive as well as of destructive nature. This model allows to obey traditional chemical-counting rules, and, based on them, bond-length contraction due to multiple bonding can be appropriately modeled. Bear in mind that the present ansatz does not contain any parameterization with respect to experimental values or data fitting. Since the Pauli kinetic energy is evaluated via the bifunctional formalism, it can, in principle, yield exactly the KS Pauli kinetic energy for an appropriately chosen deformation potential.

## 4. Materials and Methods

Similar to previously published methods [53,54], energy minimization has been performed by optimization of the respective exponents for the valence region of the participating atoms, where the electron density is represented by atom-centered, squared, real-type nodeless Slater functions of 1S and 2S-type only
(20)$$\rho \left(\vec{r}\right)=\sum_{i}\phi_{i}^{2}\left(\vec{r}-{\vec{R}}_{A}\right).$$
Those nodeless spherical Slater functions are given by [70,71]
(21)$$\phi_{i}\left(\vec{r}\right)=N_{0}r^{a_{i}}e^{-\alpha_{i}r}\;\;$$
with
(22)$$N_{0}=\sqrt{\frac{{\left(2\alpha_{i}\right)}^{2a_{i}+3}}{4\pi (2a_{i}+2)!}},$$

$a_{i}=n^{*}-1$ and $\alpha_{i}=\left(Z-s\right)/n^{*}$, where *Z* is the nuclear charge, $n^{*}$ is an effective quantum number, and *s* is the so-called shielding constant [71]. Shell concept, occupation and shielding constants have been evaluated according to the Slater rules [71], meaning that the core and valence region around each nucleus are represented by a single 1S- and 2S-type function, respectively. However, the atomic valence electron density has been restricted to closed-shell states in order to model the closed-shell state for the molecule. As mentioned above, the Slater exponents for the valence density, here $\alpha_{2S}$ in all cases, are projected to an optimization procedure in order to minimize the total energy. The optimization of exponents with respect to the energy leads to a system of non-linear equations. Therefore, the minimization procedure has to be performed iteratively.

The OF-DFT methods presented in this paper allow for the relaxation of the density as well as for a relaxation of the deformation potential, due to the interaction between the atoms. The inner part of the Pauli potential, however, obtained by the superposition of closed-shell atomic Pauli potentials, is kept fixed (at a given internuclear distance) during the optimization process. Those atomic Pauli potentials were taken from a LDA (Xonly) KS calculation performed with ADF [65] using atoms in closed-shell configurations and the QZ4P basis sets.

KS calculations for molecular PP were equally obtained at the LDA (Xonly) level using the QZ4P basis sets.

## 5. Conclusions

This study presents an ab initio, parameter-free, orbital-free implementation of density functional theory. Here, the functional value of the Pauli kinetic energy is evaluated from the density-potential pair using the bifunctional formalism. The recently introduced bifunctional formalism allows for the treatment of the functional derivative, the potential, and the electron density as two separate variables, while yielding exactly the Kohn–Sham (KS) Pauli kinetic energy when the corresponding electron density and the orbital-based KS Pauli potential are inserted. However, the bifunctional approach allows for further meaningful approximations, and thus provides a new and strategic means for the design of functionals.

This work extends the bifunctional approach beyond the recently introduced atomic fragment approximation through the help of a deformation potential that accounts for the constructive and destructive effects when two atoms approach each other. The interaction is mimicked with the help of the electron density of the participating atoms and, similar to molecular orbital (MO) theory, the interaction is taken into account with the help of a constructive or destructive combination of those densities. This approach allows study of the impact of electronic counting rules, with the help of the electron density only, and it was found that the influence of multiple bonding within the homonuclear second-row dimers is sufficiently well modeled, in the sense that the proposed model yields the shortest bond length for N${}_{2}$, followed by an increasing bond length for its next nearest neighbors (X${}_{2}$, X = C,O). Since the electron density is given as a simple monopole expansion, the present approach, thus, allows for the rationalization of chemical concepts, such as multiple bonds, together with the corresponding bond-length contraction without the necessity of introducing angular quantum numbers for the participating atoms (as they are necessary for the description of s-, p-, and d-orbitals).

## Figures and Tables

**Figure 1 molecules-26-01539-f001:**
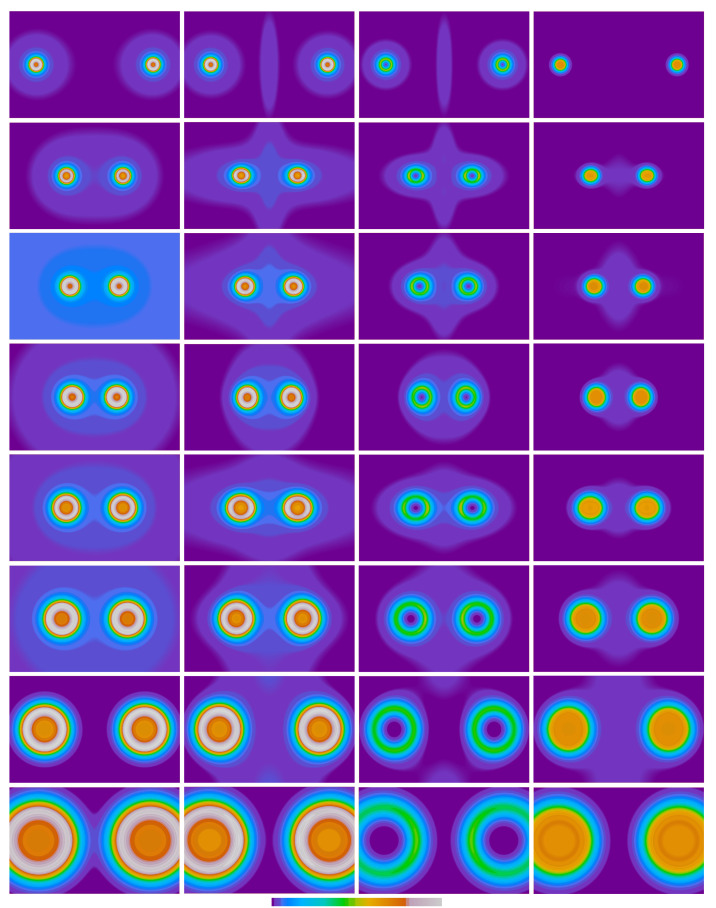
Pauli potential and components for second-row dimers. The first column depicts the Pauli potential (PP) using the bare atomic fragment approach. In the second column, the molecular PP evaluated from Kohn–Sham orbitals is compiled, together with its components $t_{P}\left(\vec{r}\right)/\rho \left(\vec{r}\right)$ and $\sum_{i}\left(\epsilon_{M}-\epsilon_{i}\right){\left|\phi_{i}\left(\vec{r}\right)\right|}^{2}/\rho \left(\vec{r}\right)$, cf. Equation (Equation 8), shown in columns three and four, respectively. First row: Ne${}_{2}$ color scale from 0.0 (blue) to 41.0 (white). Second row: F${}_{2}$ color scale from 0.0 (blue) to 36.0 (white). Third row: O${}_{2}$ color scale from 0.0 (blue) to 26.0 (white). Fourth row: N${}_{2}$ color scale from 0.0 (blue) to 18.5 (white). Fifth row: C${}_{2}$ color scale from 0.0 (blue) to 14.3 (white). Sixth row: B${}_{2}$ color scale from 0.0 (blue) to 8.9 (white). Seventh row: Be${}_{2}$ color scale from 0.0 (blue) to 5.2 (white). Eighth row: Li${}_{2}$ color scale from 0.0 (blue) to 2.4 (white). Orthoslices are shown within the range of 5 × 8 bohr for all dimers.

**Figure 2 molecules-26-01539-f002:**
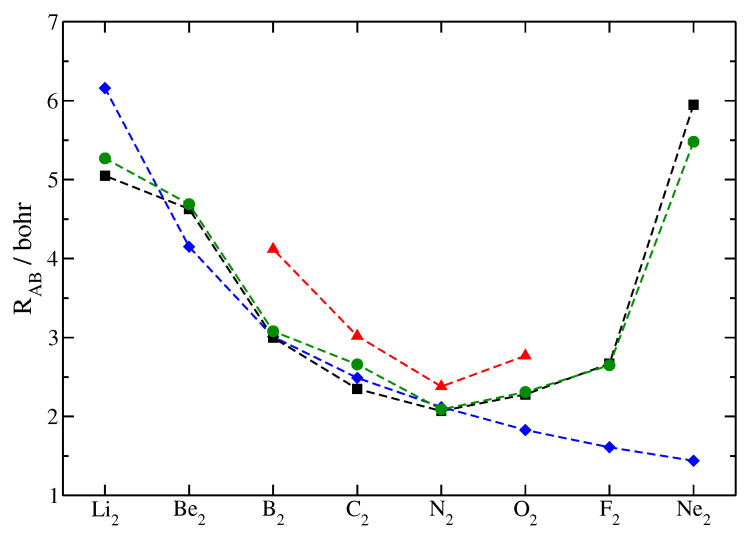
Equilibrium bond length for second-row homonuclear dimers. Black squares: Experimental values [67,68], green circles: Kohn–Sham calculations from ADF/LDA/QZ4P level, blue diamonds: OF-DFT using the bare atomic fragment approach (some of the data previously published here [54]), red triangles: OF-DFT using the deformation potential and electron counts from MO theory.

**Table 1 molecules-26-01539-t001:** Equilibrium bond distances for N${}_{2}$ from OF-DFT approaches, including various amounts of constructive versus destructive interactions between the atoms. According to molecular orbital (MO), theory N${}_{2}$ exhibits eight constructive and two destructive interaction terms. The corresponding entry is indicated in bold. The equilibrium bond distance for the bare atomic fragment approach was already published elsewhere [54].

N_2_		Constructive					
		0	2	4	6	8	10
destructive	0	2.12	2.01	1.93	1.86	1.80	1.76
	2	nb	nb	2.89	2.56	**2.38**	2.25
	4	nb	nb	nb	nb	3.28	2.93
	6	nb	nb	nb	nb	nb	/
	8	nb	nb	nb	nb	/	/
	10	nb	nb	nb	/	/	/

nb: not bound.

**Table 2 molecules-26-01539-t002:** Equilibrium bond distances for the second-row homonuclear dimers from OF-DFT using the deformation potentials in accordance with molecular orbital (MO) theory.

X_2_		Constructive				
		0	2	4	6	8
destructive	0	Ω	Li_2_ nc			
	2	nb	Be_2_ nb	B_2_ 4.12	C_2_ 3.02	N_2_ 2.38
	4	nb	nb	nb	nb	O_2_ 2.77
	6	nb	nb	nb	nb	F_2_ nb
	8	nb	nb	nb	nb	Ne_2_ nb

nb: not bound; nc: not converging.

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
