# Peer review of "Deformation Potentials: Towards a Systematic Way beyond the Atomic Fragment Approach in Orbital-Free Density Functional Theory"

_molecules, 2021, doi:10.3390/molecules26061539_

Round 1
Reviewer 1 Report
The article is a step towards a working implementation of orbital-free density functional theory (OF-DFT). Given some approximation for the Kohn-Sham potential, the main challenge of OF-DFT is to find a suitable model for the Pauli potential. The sum of Kohn-Sham and Pauli potential yields a potential that has, as its ground state, the square-root of the electron density.
Previously, the author proposed the "atomic frament approch" (AFA), where the Pauli potential is modeled as sum of the Pauli potentials for the atoms. In the manuscript, the she makes a guess about the nature of the difference between the true Pauli potential and the AFA potential. The author then investigates the predicted bond lengths for different dimers, obtaining mixed results.
It is certainly valid to make some guess for a correction to the Pauli potential, but the ansatz made in the manuscript is rather add-hoc: Atomic one-electron wavefunctions are constructed in (18) and a combined wavefunction is formed in (17) that is either bonding or anti-bonding. It is somewhat intuitive, although I would very much prefer some reasonable derivation of this form. In particular, (17) looks like the atomic one-electron wavefunctions are treated like orbitals, even though they are not.
In any case, it is an idea that is worth testing. There are, however, a some problems in the presentation:
- "S" in (17) is not defined, I think. It is probably the overlap integral.
- Are the "wavefunctions" Phi_+ and Phi_- in (17) the used like the phi_i in (9) to calculate a Pauli kinetic energy?
- The author says that the contributions need to be scaled with the number of "constructive" and "destructive" electronic interactions. What does that mean (in formulas)?
- The author states that the bond-length contraction can be reproduced without angular quantum numbers. It seems that with "angular quantum numbers", the author refers to the molecular-orbital scheme where bond formation is explained by combination of atomic orbitals (with some angular quantum number) on different atoms. However, the author's method uses, as far as I understand, the result of the molecular-orbital scheme as explicit input. Hence, I think that the statement is misleading and should be removed.
From my perspective, this scaling with the number of "constructive" and "destructive" interactions sounds like a wild mix of orbital-free with orbital-based theory. It gets some of the electronic structure recovered, but it also puts some knowledge of the electronic structure in. I cannot judge its soundness, but the goal is to find a working theory and this is a step in some direction.
The presentation is OK but the English not so much.
First page:
l4: "formalism, that" -> "formalism that"
l5: "for appropriately" -> "for an appropriately"
l7: "potential, that" -> "potential that"
l7: "account for the" -> "account the"
l8: "It is shown that ..." Please rewrite this sentence, it is very difficult to understand.
l16: "mechanics, in principle, covering all" -> "mechanics covering, in principle, all"
l17: "as failure" -> "as the failure"
l18: "structure [2,3]" -> "structures [2,3]"
l29: "and and" -> "and an"
A few more recommendations:
- All "hereby" should be replaced with "here".
- All "whereby" should be replaced with "where".
- All "notice" should be replaced with "note" (but see below).
- Commas are somewhat random. For example, there are commas before and after "that" that should not be there (3 lines after (6), line 66, 107, 154, 158, 182, 194)
- "Meaning that" is used multiple times as the start of a sentence.
- I'd remove "of course" (instead, better give a short explanation why this is obvious) and "as easily noticed by the reader".
- Asking the reading unexpectedly to "Compare", "Remember", and "Bear in mind" feels somewhat funny.
- What is "the chemical bonding curve" and, more importantly, what is its "performance"? (line 71)
- What are the "next nearest neighbors"? (line 221)
Remark: Figure 2 could easily be made more accessible for people with limited color vision by choosing different markers for the lines.
In summary, the manuscript presents an interesting idea that is certainly worth to be published. If the author fixes the description of the method, either removes or properly explains the reference to "angular quantum numbers", and fixes the English language, I recommend the manuscript for publication.
Author Response
Referee 1:
1) The article is a step towards a working implementation of orbital-free density functional theory (OF-DFT).
Given some approximation for the Kohn-Sham potential,
the main challenge of OF-DFT is to find a suitable model for the Pauli potential.
The sum of Kohn-Sham and Pauli potential yields a potential that has,
as its ground state, the square-root of the electron density.
Previously, the author proposed the "atomic fragment approach" (AFA),
where the Pauli potential is modeled as sum of the Pauli potentials for the atoms.
In the manuscript, the she makes a guess about the nature of the difference
between the true Pauli potential and the AFA potential.
The author then investigates the predicted bond lengths for different dimers, obtaining mixed results.
It is certainly valid to make some guess for a correction to the Pauli potential,
but the ansatz made in the manuscript is rather add-hoc:
Atomic one-electron wavefunctions are constructed in (18)
and a combined wavefunction is formed in (17) that is either bonding or anti-bonding.
It is somewhat intuitive, although I would very much prefer some reasonable derivation of this form.
In particular, (17) looks like the atomic one-electron wavefunctions are treated like orbitals, even though they are not.
Answer to 1): The referee is right about his assumption concerning one-electron-functions.
They are indeed the motivation for the proposed model.
In the revised version a new paragraph at the end of the Theory section
was inserted explaining this motivation in more detail
together with a comment how to make this ansatz matching the KS values.
"Additionally, the reader will notice the close connection of Eq.~\ref{eq:phi_bind_anti}
to the ansatz for classical one-electron functions in molecular orbital theory \cite{Kutzelnigg},
which by the way is the underlying reasoning for the proposed model.
However, there are two main differences rendering the
present model an approximation to the orbital-based KS PP, cf. Eq.~\ref{eq:vp}.
First of all,
the ansatz given in Eq.~\ref{eq:phi_bind_anti} is not considered to be an eigenfunction
and second, the whole deformation potential is constructed with the help
of one single atomic function, namely its electron density,
whereas in classical MO theory the ansatz is split according to the
atomic angular quantum numbers, both conceptually and computationally.
Of course, the present model can be made exact (in the sense to match the KS energy)
by searching for those functions and corresponding values $\epsilon$,
cf. Eqs.~\ref{eq:vp} and \ref{eq:tp}, obeying the necessary nodal conditions
and become the KS eigenfunctions and eigenvalues at the end of the optimization process.
In this case, however, one would finally have performed a classical KS calculation.
The point, here, is to investigate which parts of the molecular KS eigenfunctions
are the necessary ingredients for an approximate PP in order to properly model
chemical bonding.
As will be shown in the following section, in case of chemical bonding
such a mandatory ingredient seem to be the proper mixture (according to MO concept)
of constructive and destructive terms given by Eq.~\ref{eq:phi_bind_anti}
as the impact of the nodal plane given by the destructive combination
is responsible for the increasing Pauli repulsion in O$_2$, F$_2$, and Ne$_2$.
In other cases like modeling the proper atomic shell structure for atoms in their groundstate
nodal planes do not seem to play an important role,
but a proper relationship between the exponential decay and the model for the eigenvalues \cite{Finzel-2021}."
2) In any case, it is an idea that is worth testing. There are, however, a some problems in the presentation:
- "S" in (17) is not defined, I think. It is probably the overlap integral.
Answer to 2): Yes, S is the overlap integral. A note was inserted.
"and $S$ is the overlap between the functions $\Phi_A(\rv)$ and $\Phi_B(\rv)$}"
3) - Are the "wavefunctions" Phi_+ and Phi_- in (17) the used like the phi_i in (9) to calculate a Pauli kinetic energy?
4) - The author says that the contributions need to be scaled with the number
of "constructive" and "destructive" electronic interactions. What does that mean (in formulas)?
Answer to 3) and 4):
Yes, Phi_\pm are treated like phi_i in (9).
Since there is only one function for \pm, respectively, summing them up,
leads to the scaling with numbers c and d. A formula and comment has been inserted.
"\begin{equation}
\label{eq:vP_def_explizit}
v_{\textrm{P}}^{\textrm{def}}(\rv) = \frac{c \frac{1}{2} \left[\nabla \Phi_+(\rv)\right]^2 + d \frac{1}{2} \left[\nabla \Phi_-(\rv)\right]^2 - \frac{c+d}{N} t_{\textrm{W}}(\rv) }{\rho(\rv)} \;\;\; .
\end{equation}
%
where $c$ and $d$ are the number of constructive and destructive terms, respectively
and $N$ is the total number of electrons in the system. "
5) - The author states that the bond-length contraction can be reproduced without angular quantum numbers.
It seems that with "angular quantum numbers", the author refers to the molecular-orbital scheme
where bond formation is explained by combination of atomic orbitals (with some angular quantum number)
on different atoms. However, the author's method uses, as far as I understand,
the result of the molecular-orbital scheme as explicit input.
Hence, I think that the statement is misleading and should be removed.
Answer to 5):
The statement has been clarified at several positions in he text.
"The present model relies on the electron densities of the participating atoms only,
which themselves are represented by a simple monopole expansion. }
Thus, the bond length contraction can be rationalized without {\color{red} referring to } angular quantum numbers
{\color{red} of the participating atoms}. "
"like multiple bonds together with the corresponding bond length contraction
without the necessity of introducing angular quantum numbers {\color{red} for the participating atoms }
(like {\color{red} they} are necessary for the description of s-,p-, and d-orbitals). "
6) From my perspective, this scaling with the number of "constructive" and "destructive"
interactions sounds like a wild mix of orbital-free with orbital-based theory.
It gets some of the electronic structure recovered,
but it also puts some knowledge of the electronic structure in.
I cannot judge its soundness, but the goal is to find a working theory and this is a step in some direction.
Answer to 6): From my perspective, this is a reasonable well chosen ansatz.
However, I inserted a new paragraph at the end of the Theory section in order to explain it in more detail,
see answer to your first point. I hope it is more clear in the present version.
7) The presentation is OK but the English not so much.
First page:
l4: "formalism, that" -> "formalism that"
l5: "for appropriately" -> "for an appropriately"
l7: "potential, that" -> "potential that"
l7: "account for the" -> "account the"
l8: "It is shown that ..." Please rewrite this sentence, it is very difficult to understand.
l16: "mechanics, in principle, covering all" -> "mechanics covering, in principle, all"
l17: "as failure" -> "as the failure"
l18: "structure [2,3]" -> "structures [2,3]"
l29: "and and" -> "and an"
A few more recommendations:
- All "hereby" should be replaced with "here".
- All "whereby" should be replaced with "where".
- All "notice" should be replaced with "note" (but see below).
- Commas are somewhat random. For example, there are commas before and after "that" that should not be there (3 lines after (6), line 66, 107, 154, 158, 182, 194)
- "Meaning that" is used multiple times as the start of a sentence.
- I'd remove "of course" (instead, better give a short explanation why this is obvious) and "as easily noticed by the reader".
- Asking the reading unexpectedly to "Compare", "Remember", and "Bear in mind" feels somewhat funny.
- What is "the chemical bonding curve" and, more importantly, what is its "performance"? (line 71)
- What are the "next nearest neighbors"? (line 221)
Answer to 7):
Concerning your suggestions about the English language,
I have taken all your advice and changed the text accordingly,
except for line 18. Here, atomic shell structure is meant as a proper noun
and not to indicate that there are several atomic shell structures.
However, in order to follow your suggestion and make the meaning more clear
the text has been changed to "the atomic shell structure."
I have replaced all "hereby", "whereby", and "notice" with
"here", "where", and "note", respectively.
Commas have been corrected according to your advice
and I also tried to spare "Compare", "Remember", and "Bear in mind" more often.
Casual language concerning the "chemical bonding curve" its "performance"
and the next nearest neighbors to N$_2$ have been omitted, rephrased or clarified.
8) Remark: Figure 2 could easily be made more accessible for people with limited color vision
by choosing different markers for the lines.
Answer to 8): The figure has been changed.
9) In summary, the manuscript presents an interesting idea that is certainly worth to be published.
If the author fixes the description of the method,
either removes or properly explains the reference to "angular quantum numbers",
and fixes the English language, I recommend the manuscript for publication.
Answer to 9): Your requests have been addressed in the revised version.
Reviewer 2 Report
The manuscript is dedicated to contribute into the orbital-free density functional theory on the idea of constructive versus destructive interaction of atomic electron densities and its performance with the respect to equilibrium bond lengths. The author claims the bifunctional approach provides a new and strategic way in the design of functionals. Thus, the author must update citations or add more precise explanations:
- There should be listed what other ways are used for design of new functional. Which information does allow to claim the OF-DFT is strategic way for the design?
- On the page 4 there are descriptions of constructive and destructive interaction without the citations: if it is the new idea there should be more description what does it mean; if it is based on other works they must be cited. Especially it applies to for the equations (13) and (17).
- It is unclear why author neglects the Full-Configuration-Interaction (FCI) level study. The functional must be compared with FCI level results. Moreover, the studies are for two atoms only.
- The introduction should explain what the application should be use the OF-DFT approach. I don’t think it can be suitable for polyene or carotenoids dark states studies. Thus, the limitations must be provided.
- The journal title is “Molecules”. The reader would expect the usefulness for large molecules and it must be discussed.
- The page 5 text is massy and it must be organized in more clear way. It is difficult to follow references from text to Figure 1 subfigures lines.
- Figure 1 caption must be updated in more precise way: what does it mean “value ranges”?
- Table 1. As it is said in text there were results with the other atoms. They could be presented in Supplementary for the review.
- The manuscript has grammar errors and it must be checked. Some errors:
Line 7: „..that takes into account for the interaction..“ „..that takes into account the interaction..“
Line 122: combinations – singular form, should be plural;
Line 161: „the the“ repeatance;
Line 172: „extend“, i assume the author meant extent
Line 202: „..were eqaully be obtained..“; either „ were equally obtained“ or „can equally be obtained“
Line 219 to 220: „.. can sufficiently well modeled“ either „can be ..“ or „ ..is sufficiently well modeled“; either missing verb or wrong choice;
Line 224: „..like the are necessary..“, probably missing y -> they, otherwise wrong composition.
Figure 1: „Eights row“: „Eight row“
Author Response
Referee 2:
1) The manuscript is dedicated to contribute into the orbital-free density functional theory
on the idea of constructive versus destructive interaction of atomic electron densities
and its performance with the respect to equilibrium bond lengths.
The author claims the bifunctional approach provides a new and strategic way in the design of functionals.
Thus, the author must update citations or add more precise explanations:
•There should be listed what other ways are used for design of new functional.
Which information does allow to claim the OF-DFT is strategic way for the design?
Answer to 1): Other approaches for the design of the kinetic energy are given in the Introduction,
see for example second and third paragraph, together with the corresponding citations.
The recent approach is strategic because it relies on the bifunctional reformulation,
which provides exact access to non-interacting kinetic energy. The method is explained,
references to previous work are given.
2)• On the page 4 there are descriptions of constructive and destructive interaction
without the citations: if it is the new idea there should be more description what does it mean;
if it is based on other works they must be cited.
Especially it applies to for the equations (13) and (17).
Answer to 2):
A new paragraph explaining the approach in more detail was inserted.
"%
{\color{red}
\begin{equation}
\label{eq:vP_def_explizit}
v_{\textrm{P}}^{\textrm{def}}(\rv) = \frac{c \frac{1}{2} \left[\nabla \Phi_+(\rv)\right]^2 + d \frac{1}{2} \left[\nabla \Phi_-(\rv)\right]^2 - \frac{c+d}{N} t_{\textrm{W}}(\rv) }{\rho(\rv)} \;\;\; .
\end{equation}
%
where $c$ and $d$ are the number of constructive and destructive terms, respectively
and $N$ is the total number of electrons in the system.
Note} that, this approach allows to treat electron counting rules, e.g.
originating from a MO graph,
within an OF-DFT method
as well as for the separate treatment of core and valence electrons.
{\color{red}
The input of $c$ and $d$ does, of course, require some knowledge from electronic structure theory
and so does the concept of separation between core and valence electrons.
Such concepts, however, are at the basis of almost every chemical reasonings and
find their application outside quantum chemistry as well, e.g. the commonly used octet rule
which serves to explain the vast majority of molecular compositions.
Additionally, the reader will notice the close connection of Eq.~\ref{eq:phi_bind_anti}
to the ansatz for classical one-electron functions in molecular orbital theory \cite{Kutzelnigg},
which by the way is the underlying reasoning for the proposed model.
However, there are two main differences rendering the
present model an approximation to the orbital-based KS PP, cf. Eq.~\ref{eq:vp}.
First of all,
the ansatz given in Eq.~\ref{eq:phi_bind_anti} is not considered to be an eigenfunction
and second, the whole deformation potential is constructed with the help
of one single atomic function, namely its electron density,
whereas in classical MO theory the ansatz is split according to the
atomic angular quantum numbers, both conceptually and computationally.
Of course, the present model can be made exact (in the sense to match the KS energy)
by searching for those functions and corresponding values $\epsilon$,
cf. Eqs.~\ref{eq:vp} and \ref{eq:tp}, obeying the necessary nodal conditions
and become the KS eigenfunctions and eigenvalues at the end of the optimization process.
In this case, however, one would finally have performed a classical KS calculation.
The point, here, is to investigate which parts of the molecular KS eigenfunctions
are the necessary ingredients for an approximate PP in order to properly model
chemical bonding.
As will be shown in the following section, in case of chemical bonding
such a mandatory ingredient seem to be the proper mixture (according to MO concept)
of constructive and destructive terms given by Eq.~\ref{eq:phi_bind_anti}
as the impact of the nodal plane given by the destructive combination
is responsible for the increasing Pauli repulsion in O$_2$, F$_2$, and Ne$_2$.
In other cases like modeling the proper atomic shell structure for atoms in their groundstate
nodal planes do not seem to play an important role,
but a proper relationship between the exponential decay and the model for the eigenvalues \cite{Finzel-2021}. "
3) • It is unclear why author neglects the Full-Configuration-Interaction (FCI) level study.
The functional must be compared with FCI level results. Moreover, the studies are for two atoms only.
Answer to 3):
This is not a study aiming to approximate the exchange-correlation functional, where
comparison to an explicitely correlated post-HF method is of course of interest.
Here, the kinetic energy is approximated and therefore, comparison is made to the
corresponding KS data as well as to experimental values.
Yes, this study is based on dimers.
4) • The introduction should explain what the application should be use the OF-DFT approach.
I don’t think it can be suitable for polyene or carotenoids dark states studies.
Thus, the limitations must be provided.
Answer to 4):
As mentioned in answer 3), the point here is to approximate the kinetic energy
(not the exchange-correlation part of the electron-electron repulsion),
which is a considerably bigger challenge due to the different amount in the
order of magnitude with respect to the total energy.
Therefore, the study presents a new idea with application on homonuclear dimers.
Carotenoids dark state studies are of course an interesting topic left for the future.
5) • The journal title is “Molecules”.
The reader would expect the usefulness for large molecules and it must be discussed.
Answer to 5):
As mentioned in 3) and 4), the target approximating T_s is still a challenge.
This work presents a new approach. From my perspective it seems natural
if not mandatory to start with applications on small systems before targeting
large molecules, as one would probably not capture all effects at once and might run
into circling reasoning or data fitting.
However, the referee is of course right, that large molecules are of interest.
The work was sent to the SI in the honor of Linus Pauling. On the website it was stated that:
"It is with this situation of affairs in mind that it is our great pleasure to invite our colleagues
working in the broad field of structure and chemical bonding to participate to this
Linus Pauling commemorative issue. We envisage contributions from several disciplines
having chemical structure (geometrical and electronic) and its understanding,
rationalization and prevision through chemical bonding as one of the key ingredients.
Scientists developing or making use of either experimental or theoretical methods
and scientists analyzing chemical bonding within formalisms based on Hilbert space entities
or defined in the real space (quantum chemical topological methods) are highly welcome
to contribute to this issue. Papers on recent developments in the valence bond theory
and models are also eagerly awaited, as well as on any experimental or theoretical progress
able to shed light on chemical bonding issues. Applications to challenging structural
and chemical bonding cases, including those occurring under extreme conditions,
such as high pressure, are also highly welcome. The diversity of contributions,
coming from several fields of science and from scientists with different expertise,
will be a faithful representation of the diversity as well as of the profound unity
and interrelationship of Linus Pauling interests. He remains an inspiration to all of us
and we are pleased to invite you to submit a publication for this Special Issue."
From my perspective, the work fits the issue.
6) • The page 5 text is massy and it must be organized in more clear way.
It is difficult to follow references from text to Figure 1 subfigures lines.
• Figure 1 caption must be updated in more precise way: what does it mean “value ranges”?
• Table 1. As it is said in text there were results with the other atoms.
They could be presented in Supplementary for the review.
Answer to 6):
Figure caption was updated and a color scale inserted for better understanding.
However, there is no supplementary material.
7) • The manuscript has grammar errors and it must be checked. Some errors:
Line 7: „..that takes into account for the interaction..“ „..that takes into account the interaction..“
Line 122: combinations – singular form, should be plural;
Line 161: „the the“ repeatance;
Line 172: „extend“, i assume the author meant extent
Line 202: „..were eqaully be obtained..“; either „ were equally obtained“ or „can equally be obtained“
Line 219 to 220: „.. can sufficiently well modeled“ either „can be ..“ or „ ..is sufficiently well modeled“; either missing verb or wrong choice;
Line 224: „..like the are necessary..“, probably missing y -> they, otherwise wrong composition.
Figure 1: „Eights row“: „Eight row“
Answer to 7):
The proposed corrections with respect to English grammar and spelling
have fully been taken into account in the revised version.
Reviewer 3 Report
While the existence of a universal density functional for the kinetic energy is known, finding a good approximation for it is still a challenge. The present manuscript explores a hybrid approach to find a solution. The method is also connected to descriptors of the chemical bond, and can be published in the special issue for which it is submitted.
The manuscript can be accepted in its present form. However, I would like to submit the following suggestions to the author, to whom the decision to take them into account, or not, should be left.
- I think that it would be useful to extend the last paragraph, explaining in more detail the steps taken to obtain the results.
- The problem of the density functionals for the kinetic energy is not obtaining a reasonable value for the kinetic energy, but for energy difference. It might be interesting to insist on the latter. (I realize that - in an indirect way - this is addressed by the results presented in Tab. 1.)
Author Response
Referee 3:
1) While the existence of a universal density functional for the kinetic energy is known,
finding a good approximation for it is still a challenge.
The present manuscript explores a hybrid approach to find a solution.
The method is also connected to descriptors of the chemical bond,
and can be published in the special issue for which it is submitted.
The manuscript can be accepted in its present form.
However, I would like to submit the following suggestions to the author,
to whom the decision to take them into account, or not, should be left.
1. I think that it would be useful to extend the last paragraph,
explaining in more detail the steps taken to obtain the results.
Answer to 1):
The last paragraph of the Theory section was enlarged,
explaining in more detail the motivation behind the concept as
well as the explicit formula how to obtain the deformation potential.
%
{\color{red}
\begin{equation}
\label{eq:vP_def_explizit}
v_{\textrm{P}}^{\textrm{def}}(\rv) = \frac{c \frac{1}{2} \left[\nabla \Phi_+(\rv)\right]^2 + d \frac{1}{2} \left[\nabla \Phi_-(\rv)\right]^2 - \frac{c+d}{N} t_{\textrm{W}}(\rv) }{\rho(\rv)} \;\;\; .
\end{equation}
%
where $c$ and $d$ are the number of constructive and destructive terms, respectively
and $N$ is the total number of electrons in the system.
Note} that, this approach allows to treat electron counting rules, e.g.
originating from a MO graph,
within an OF-DFT method
as well as for the separate treatment of core and valence electrons.
{\color{red}
The input of $c$ and $d$ does, of course, require some knowledge from electronic structure theory
and so does the concept of separation between core and valence electrons.
Such concepts, however, are at the basis of almost every chemical reasonings and
find their application outside quantum chemistry as well, e.g. the commonly used octet rule
which serves to explain the vast majority of molecular compositions.
Additionally, the reader will notice the close connection of Eq.~\ref{eq:phi_bind_anti}
to the ansatz for classical one-electron functions in molecular orbital theory \cite{Kutzelnigg},
which by the way is the underlying reasoning for the proposed model.
However, there are two main differences rendering the
present model an approximation to the orbital-based KS PP, cf. Eq.~\ref{eq:vp}.
First of all,
the ansatz given in Eq.~\ref{eq:phi_bind_anti} is not considered to be an eigenfunction
and second, the whole deformation potential is constructed with the help
of one single atomic function, namely its electron density,
whereas in classical MO theory the ansatz is split according to the
atomic angular quantum numbers, both conceptually and computationally.
Of course, the present model can be made exact (in the sense to match the KS energy)
by searching for those functions and corresponding values $\epsilon$,
cf. Eqs.~\ref{eq:vp} and \ref{eq:tp}, obeying the necessary nodal conditions
and become the KS eigenfunctions and eigenvalues at the end of the optimization process.
In this case, however, one would finally have performed a classical KS calculation.
The point, here, is to investigate which parts of the molecular KS eigenfunctions
are the necessary ingredients for an approximate PP in order to properly model
chemical bonding.
As will be shown in the following section, in case of chemical bonding
such a mandatory ingredient seem to be the proper mixture (according to MO concept)
of constructive and destructive terms given by Eq.~\ref{eq:phi_bind_anti}
as the impact of the nodal plane given by the destructive combination
is responsible for the increasing Pauli repulsion in O$_2$, F$_2$, and Ne$_2$.
In other cases like modeling the proper atomic shell structure for atoms in their groundstate
nodal planes do not seem to play an important role,
but a proper relationship between the exponential decay and the model for the eigenvalues \cite{Finzel-2021}.
2) 2. The problem of the density functionals for the kinetic energy
is not obtaining a reasonable value for the kinetic energy,
but for energy difference. It might be interesting to insist on the latter.
(I realize that - in an indirect way - this is addressed by the results presented in Tab. 1.)
Answer to 2):
For sure, the referee is right.
Indeed this issue has been addressed by the data in Table 1.
However, following the referees advice I inserted a few more paragraphs
addressing this issue.
"{\color{red}
Whereas, it is relatively easy to obtain the KS Pauli kinetic energy for a given system of interest,
and consequently its total KS energy,
e.g. simply by inverting the KS equations and defining the deformation potential as the difference between the
KS Pauli potential
and the bare atomic fragment approach,
it is of great difficulty to predict the performance of a given approximate deformation potential in advance.
The challenge, here, is not only to yield good energetic agreement for a special case,
but to obtain reasonable energy differences, e.g
for a molecule at various geometries.
The reasonable description of chemical bonding,}
thus, lies in the choice for an appropriate deformation potential $v_{\textrm{P}}^{\textrm{def}}(\rv)$.
"
"{\color{red}
As mentioned in the Theory section obtaining a reasonable OF-DFT kinetic energy
for a special case can be done numerically in a straightforward way by inversion of the KS equations.
Equally, one could think of a certain model with adjustable parameter in order to obtain the desired match
(like it was frequently done with the von Weizs\"acker correction to the Thomas-Fermi theory \cite{Yonei-Tomishima-1965}).
In contrast to single point matches, the design of kinetic energy functionals
yielding appropriate energy differences, e.g. for varying nuclear coordinates, is still challenging.
The model proposed in this work does not only yield reasonable energy differences
for a chosen deformation potential with fixed number for $c$ and $d$, and thus,
yielding bound atoms with reasonable bond distances,
but also reasonable results among different ansatzes (using different numbers for $c$ and $d$).
Changes in the bond length (the minima of the respective bonding curves)
follow the reasoning from traditional electronic structure theory.
}"
Round 2
Reviewer 2 Report
The manuscript was improved. As the author mentioned it wasn't made in all cases. However, now it is much better and can be published. The English changes were maid.